# Exploring the effects of quercetin-added pancreatic diet on metabolic homeostasis in dogs via metabolomics

Xiao-Wan Liu[1,2☉], Yao-hui Zhang[1,3☉], Li Xu[1,2☉], Jia-Bao Xing[4], Zhou-xiang Wang[1,5], Man-li Hu[1,6], Yun Chen[1,7], Zhi-li Qi[8], Yi Ding[9], Xin Zhang[1,6], Ming-Xing Ding[9], Xiao-Jing Zhang[1,5], Juan Wan[1,6]*

1 Gannan Innovation and Translational Medicine Research Institute, First Affiliated Hospital, Gannan Medical University, Ganzhou, China, 2 School of Public Health and Health Management, Gannan Medical University, Ganzhou, China, 3 School of Basic Medicine, Gannan Medical University, Ganzhou, China, 4 Jiangxi Huichong Technology Co., Ltd., Ganzhou, China, 5 School of Basic Medicine, Wuhan University, Wuhan, China, 6 Key Laboratory of Prevention and Treatment of Cardiovascular and Cerebrovascular Diseases, Ministry of Education, Gannan Medical University, Ganzhou, China, 7 Huanggang Institute of Translational Medicine, Huanggang, China, 8 College of Animal Science and Technology, Huazhong Agricultural University, Wuhan, China, 9 College of Veterinary Medicine, Huazhong Agricultural University, Wuhan, China

☉ These authors contributed equally to this work and share first authorship.
* wanjuan@gmu.edu.cn

## Abstract

### Objective

To investigate the role of quercetin-added pancreatic prescription food in regulating metabolic homeostasis in dogs.

### Methods

The experimental dogs were divided into a control diet group and a prescription diet group. The control group was fed regular food, while the prescription group was fed pancreatic prescription food (3.9 g of quercetin was added in per 1 kg of food) for 8 weeks. Canine physical examination, complete blood count, and serum biochemical tests were conducted at 0 w, 4 w, and 8 w. Non-targeted metabolomics tests were performed using plasma samples at 0 w and 8 w.

### Results

Dogs that received a quercetin-added pancreatic diet supplemented with quercetin showed no changes in the body weight, fasting blood glucose, body condition score, the indexes of whole blood program of red blood cells, white blood cells and platelets, and most blood biochemical indexes, but increased lipase levels in plasma at 8 w. Quercetin significant improved in metabolic homeostasis, especially in fatty acid, amino acid, and bile acid metabolism. Untargeted metabolomics analysis revealed that quercetin activates ABC transport and arginine/proline pathways, suggesting potential benefits for pancreatitis in large animals, while maintaining comparable safety parameters.

**Data availability statement:** All relevant data are within the manuscript and its Supporting information files.

**Funding:** This work was supported by the National Science Foundation of China (Grant No. 82170595 and 81970070 to X-JZ), the Henan Charity General Federation-Hepatobiliary Foundation of Henan Charity General Federation (Grant No. GDXZ2022008 to XZ, GDXZ2022009 to WJ), Huazhong Agricultural University-Agricultural Genomics Institute at Shenzhen, Cooperation Fund (No. SZYJY2022008 to ZQ), National special fund for research and development (No. YFD051009 to M-XD). Gannan Innovation and Transformation Medical Research Institute, 4# building, Phase II, high level talents Park, Zhanggong high-tech zone, Ganzhou City, Jiangxi Province.

**Competing interests:** We would like to submit the enclosed manuscript entitled "Exploring the Effects of Quercetin-added Pancreatic Diet on Metabolic Homeostasis in Dogs via Metabolomics". No conflict of interest exits in the submission of this manuscript, and manuscript is approved by all authors for publication. I would like to declare on behalf of my co-authors that the work described was original research that has not been published previously, and not under consideration for publication elsewhere, in whole or in part. All the authors listed have approved the manuscript that is enclosed.

## Conclusions

Quercetin-added prescription food enhances fatty acid and amino acid metabolism, demonstrating its potential to promote pancreatic function and sustain metabolic homeostasis.

## 1. Introduction

Pancreatitis is a potentially devastating, life-threatening condition that affects as many as 80,000 people in the United States annually [1]. Acute pancreatitis is a relatively common clinical condition hallmarked by unregulated trypsin activity within the pancreatic acinar cell, leading to pancreatic autodigestion and parenchymal inflammation. In contrast, chronic pancreatitis is a progressive fibro inflammatory disease characterized by irreversible loss of the pancreatic parenchyma and subsequent functional insufficiency. Canine pancreatitis has been reported to manifest a similar condition in humans [2]. Dogs with severe acute pancreatitis are typically displayed an acute onset of anorexia, weakness, vomiting, diarrhoea and/or abdominal pain [3,4]. Chronic pancreatitis in dogs is typically associated with intermittent clinical signs that are less specific and milder than acute pancreatitis [5–7]. Thus, the study of canine pancreatitis helps to address the lack of understanding of its pathophysiological mechanism and provides effective treatment strategies for human pancreatitis.

The development of pancreatitis is the result of various mechanisms, including premature activation of digestive enzymes within pancreatic acinar cells, activation of inflammatory pathways, altered pancreatic microcirculation, oxidative stress, activation of the complement system [8,9]. Clinical treatment of canine pancreatitis usually contains intravenous injections, antiemetics, analgesia, antibacterial and anti-inflammatory, nutritional supplements and other measures [10]. At present, the incidence of canine pancreatitis is on the rise, leading to increased economic pressure to pet owners [11]. Therefore, it is of significant clinical importance and social value to develop effective pancreatitis treatment drugs for pancreatitis or prescription foods for pets with pancreatic issues.

Quercetin is one of the most widely distributed and most extensively studied flavonoid that are widely present in the human diet. It is found to play a role in protecting pancreatic related disease, such as pancreatitis, insulin dysfunction, and pancreatic cancer. Quercetin protected against intestinal barrier disruption and inflammation in acute pancreatitis [12]. Quercetin strengthened the survival processes and the secretory capacity of beta cells through several mechanisms, including inhibition of NF-κB signaling, nitric oxide generation, reactive oxygen species levels, increased mitochondrial bioenergetic function, and stimulated pathways of insulin secretion (e.g., PLC/PKC cAMP/PKA, and/or mTOR signaling) [13]. Quercetin supplementation is considered a promising pancreatic-protect compound and to maintain human and other animal health.

In this study, the dogs were fed with an independently developed pancreatic prescription food with quercetin-added, and canine physiological performance, complete blood count, blood biochemical index and plasma metabolomics were investigated to evaluate the regulatory effect of the prescription food on pancreatic function and metabolic homeostasis.

## 2. Materials and methods

### 2.1. Ethics statement

This research was sanctioned by the Biomedical Research Ethics Committee of Gannan Medical University, reference number 2021335, in compliance with the pertinent regulations

outlined in the Guiding Opinions on the Humane Treatment of Laboratory Animals, document No. 398 issued by the Ministry of Science and Technology of the People's Republic of China, National Science and Technology Department (2006). A total of ten healthy field dogs were recruited from rural Ganzhou between October 2021 and October 2023 for this investigation. Prior to the commencement of the study, written consent was secured from the dogs' owners, and the opportunity for participation was communicated to the owners of dogs identified as potential candidates. All participating dogs underwent a comprehensive physical examination, systemic blood pressure assessment via the Doppler method, electrocardiography, and echocardiography. The study did not involve anesthesia or euthanasia, and no dogs experienced mortality. Following the conclusion of the experiment, all dogs were maintained in a standard animal housing facility for one month before being returned to their owners.

## 2.2. Prescription food preparation

The formula of canine pancreatic prescription food was designed by referencing the national Standard (GB/T 31216-2014), Full-price Pet Food-Dog Food and American Association of Feed Control Officials (AAFCO), and considering the pathogenesis and nutritional requirements of pancreatitis. The raw materials and additives used in the formula were in line with the provisions of the Feed Raw Materials Catalogue and Feed Additives Catalogue. The preparation process of prescription food was as follows. Firstly, the raw material with large particles was crushed, sieved and weighed according to the formula, stirred evenly in the mixer. Next, the mixed material was puffed and granulated in the expander. Then the expanded granular material was dried at 65 °C for 4 h to reduce the moisture content to 5%–10%. Finally, the dried prescription food was vacuum bagged. The crude protein, crude fat, crude fiber, ash, moisture, calcium, total phosphorus and energy of the pancreas prescription diet were determined referring to national standards or general method.

## 2.3. Experimental animals

Ten healthy field dogs were recruited from a clinical practice. The body weight was 8~10 kg, the age was 1~5 years old with no history of disease. Routine in vitro and in vivo deinsectization and vaccine immunization were completed before the trial. This project was approved by the Biomedical Research Ethics Committee of Gannan Medical College (No. 2021335). The dogs were randomly divided into control diet group (n = 4) and a pancreas prescription diet group (n = 6). The control diet group was fed with commercial full price compound food for experimental dogs. The prescription diet group were fed with canine pancreatic prescription food, which was jointly developed by Gannan Institute of Innovation and Translational Medicine and Jiangxi Huichong Technology Co., LTD. All dogs underwent a 2-week food adaptation period before formal feeding. The daily feeding amount was calculated according to the formula: MER/ME, MER = $BW_{kg}^{0.75} \times 70 \times$ life stage factor.

## 2.4. Canine physical examination

The experimental dogs were fed twice a day (at 9:00 am and 6:00 pm) for 8 weeks. Their daily food intake was recorded during the experiment. Physical examinations including body weight, fasting blood glucose levels, and body condition score (BCS) were performed at 0 w, 4 w, and 8 w. The dogs were weighed after fasting overnight, and their body condition was independently scored by two researchers.

## 2.5. Complete blood count and serum biochemical tests

The blood samples were collected for a complete blood count and blood biochemistry examination at 0 w, 4 w, and 8 w. Blood samples at 0 w and 8 w were also applied metabolically test. Five milliliters of whole blood were collected from the distal cephalic vein. The fasting blood glucose was tested on Blood glucose meter (AccU-Chek Active, Roche, USA) with 20 μL of blood. The blood count was applied on five-category blood count apparatus for animals (Abaxis VetScan HM5, USA) using 1 mL EDTA anticoagulant blood. The remaining blood was stored in a heparin lithium anticoagulant tube and centrifuged at 1500 g for 10 min to obtain upper plasma for biochemical test and metabolomics detection. The biochemical detection was applied on Automatic biochemical analyzer (Hitachi 3110, Japan). Complete blood count and blood biochemical tests were performed by the medical Laboratory of WellAnimal Test (Wuhan) Co., LTD.

## 2.6. Untargeted metabolomics

The plasma samples for metabolomics detection were transferred to EP tubes and immediately frozen in liquid nitrogen. Before testing, the plasma samples were thawed slowly at 4 °C, and 100 mL of each sample was vortically mixed with 400 mL of pre-cooled methanol/acetonitrile/aqueous solution (2:2:1, V/V). The samples were then ultrasonically treated at 4 °C for 30 min, and stood at −20 °C for 10 min. After that, the samples were centrifuged at 4 °C at 14000 g for 15 min in a low temperature high speed centrifuge (Eppendorf 5430R) to remove the residual proteins, and the supernatant was dried by vacuum. The dried samples were redissolved in acetonitrile solution, and the supernatant was collected by centrifugation for untargeted metabolome sequencing. Untargeted metabolome sequencing was performed by high resolution liquid chromatography (Agilent 1290 Infinity, Agilent, USA) combined with the flight time mass spectrometry (AB Triple TOF 6600, AB Sciex, USA). XCMS software was used for peak alignment, retention time correction and peak area extraction of mass spectrometry, and finally for qualitative and quantitative analysis of metabolites.

## 2.7. Bioinformatics analysis

Metabolite identification and relative abundance calculations were carried out with metabolomics original results. The metabolite abundance matrix was imported into Ropls software (version No. 1.26.0) for principal component analysis to assess the consistency distribution of samples in different groups. The multivariate statistical OPLS-DA model was used to analyze and screen the metabolites with different abundance. The HMDB database was used to exclude endogenous synthetic metabolites, and then KEGG pathway enrichment analysis was conducted for metabolites with different abundance. Meanwhile, the metabolic spectrum trend of each KEGG metabolic pathway was determined based on the proportion of all metabolites up- and down-regulated in different groups.

## 2.8. Statistical analysis

SPSS 21.0 software was used to analyze the difference of blood count and blood biochemical indexes between groups. All data are expressed as mean ± SD. When the data were in line with normal distribution, independent T test was used to compare the difference between the two groups. The non-parametric Mann-Whitney U test was used when the distribution does not conform to a normal distribution. When $P < 0.05$, the difference between groups was considered statistically significant.

## 3. Results

### 3.1. Nutrition component of pancreatic prescription food

The crude protein, crude fat, crude fiber, ash, moisture, calcium, total phosphorus and energy values of the food were determined referring to national standards or general method. The detail information of each nutrient component per 100 g of dry matter in both control commercial food and pancreatic prescription food were is presented in Table 1. There were 3.1 g of crude fat and 4 g of crude fiber per 100 g dry matter of pancreatic prescription food, while 12.06 g of crude fat and 9 g of crude fiber in that of control food. The added amount of quercetin was 3.6 g per 100 g of dry matter in the pancreatic prescription food. The metabolic energy of the prescription food (3126.16 kcal/kg) is close to control food (3152.09 kcal/kg). These results illustrated that the formulated pancreatic prescription food has a portrait of low fat, highly digestible and energy balanced.

### 3.2. Effect of quercetin-added pancreatic prescription diet on physiological indexes and complete blood count in dogs

The experimental dogs were fed with control or quercetin-added pancreatic prescription diet, and showed no abnormality in daily observation. Physical examination, complete blood count and blood biochemistry were performed at 0 w, 4 w and 8 w (Fig 1A). The body weight, fasting blood glucose, BCS and other physical indicators of experimental dogs were analyzed and exhibited no difference in the two groups during the experiment (Fig 1B and C). The indexes of whole blood program of red blood cells, white blood cells and platelets showed no change between pancreatic prescription group and control group (Fig 1D), which indicates the quercetin-added pancreatic prescription diet has no toxic on blood system.

### 3.3. Effect of quercetin-added pancreatic prescription diet on blood biochemical indexes in dogs

To fully investigate the safety of the quercetin-added pancreatic prescription diet on key organs, the biochemistry parameters were systematically tested. The lipase levels in plasma were significantly decreased in pancreatic prescription-diet dogs at 8 w. No changes were found in the levels of α-amylase between two groups during the experiment. The levels of total protein (TP), albumin (ALB), alanine aminotransferase (ALT), aspartate aminotransferase (AST), alkaline phosphatase (ALP), triglyceride (TG), total cholesterol (TC), α-amylase (α-AMY), creatine kinase (CK), creatinine (CRE), blood urea nitrogen (BUN), calcium ion ($Ca^{2+}$), potassium ion ($K^{2+}$), and sodium ion ($Na^{2+}$) between pancreatic prescription group and control group showed no obvious difference (Fig 2). These biochemical data indicated that the

Table 1. Nutritional components of control diet and pancreatic prescription diet.

| Nutritional components | Control diet | Prescription diet |
|---|---|---|
| Crude protein (g) | 18.95 | 21.48 |
| Crude ash (g) | 9.61 | 6.1 |
| Crude fat (g) | 12.06 | 3.1 |
| Crude fibre (g) | 9 | 4 |
| Water (g) | 7.65 | 10 |
| Total carbohydrate (g) | 51.73 | 59.32 |
| Quercetin (g) | 0 | 3.6 |
| Metabolic energy (kcal/kg) | 3152.09 | 3126.16 |

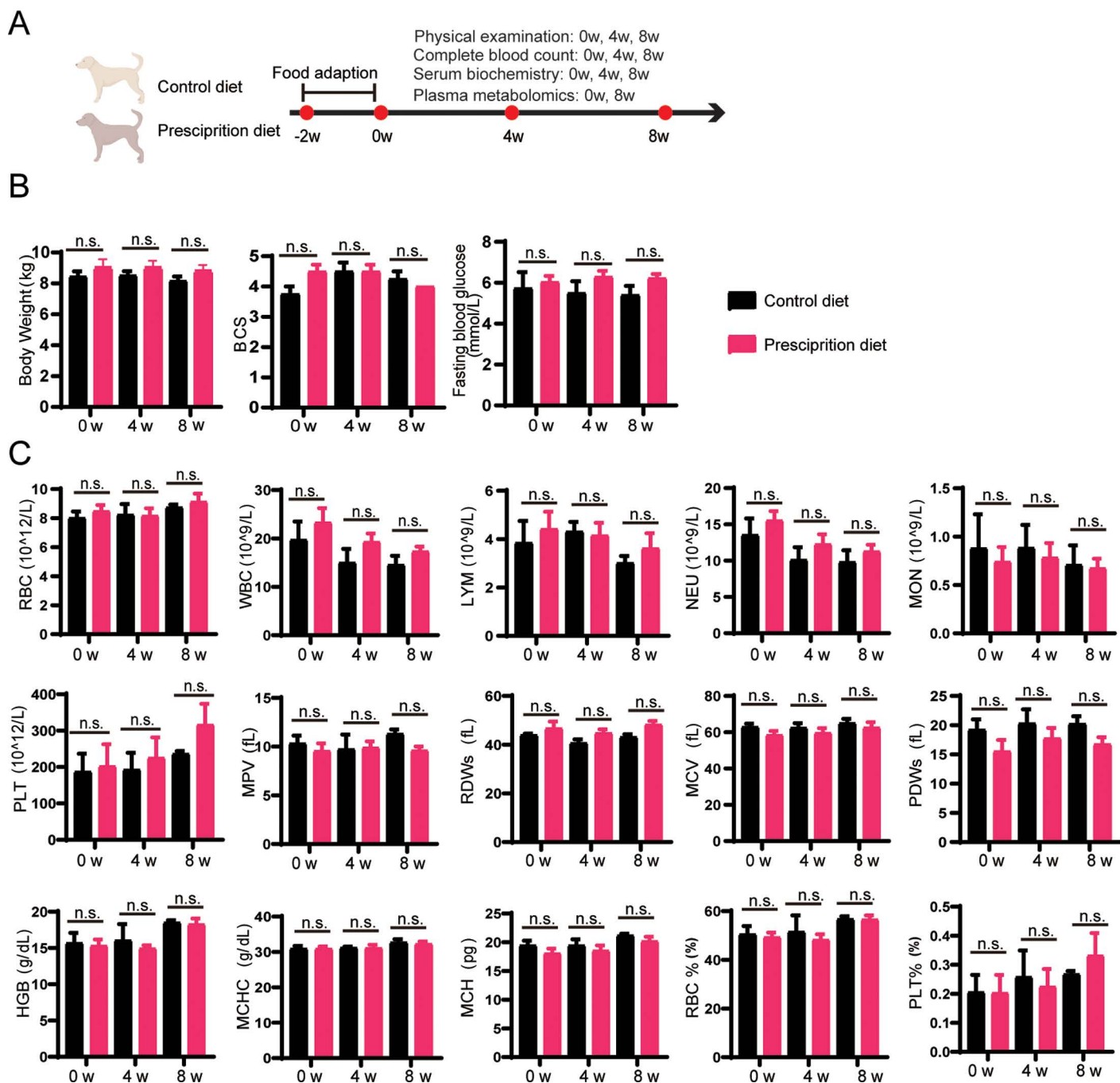

**Fig 1. The analysis of physiological indexes and complete blood count in dogs fed with quercetin-added pancreatic prescription diet.** (A) Schematic diagram of the experimental design process. Dogs were randomly divided into two groups, and then fed with a quercetin-added pancreatic prescription diet or a control diet for 8 weeks after a 2-week adaptation period, respectively. The dogs were evaluated by physical examination, laboratorial examination at 0w, 4w and 8w. (B) Analysis of body weight, body condition score (BCS) and fasting blood glucose in the prescription diet group and control diet group at 0w, 4w and 8w. n = 6 in prescription diet group, and n = 4 in control diet group. (C) Whole blood parameter analysis of dogs fed with prescription diet and control diet. n = 6 in prescription diet group, and n = 4 in control diet group. RBC, red blood cell; WBC, white blood cell; LYM, lymphocytes; MON, monocytes; NEU, neutrophils; PLT, platelets; MPV, mean platelet volume; RDWs, red cell distribution width; MCV, mean corpuscular volume; PDWs, platelet distribution width; HGB, hemoglobin; MCHC, mean corpuscular hemoglobin concentration; MCH, mean corpuscular hemoglobin; RBC%, the percent of red blood cell in blood cells; PLT%, the percent of platelets in blood cells. n.s. represents a P value greater than 0.05 (not statistically significant).

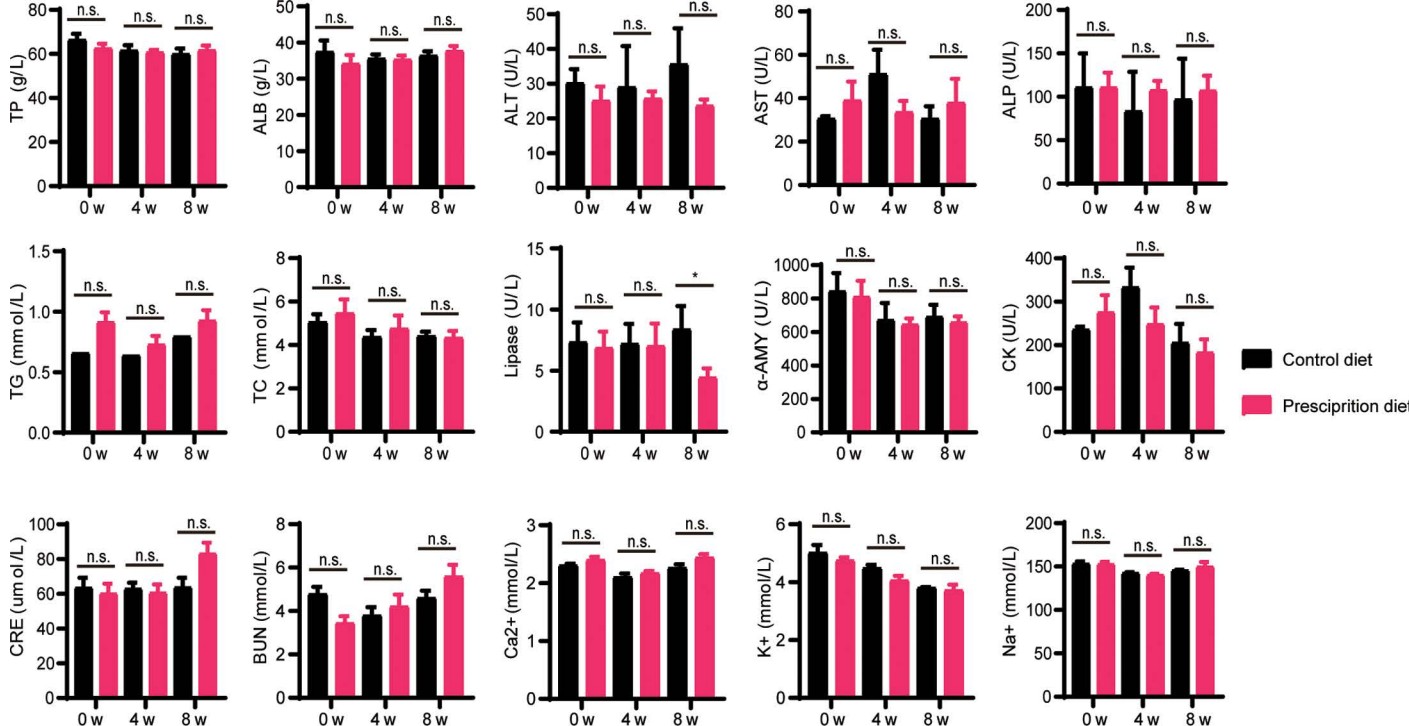

**Fig 2. The analysis of blood biochemical indexes in dogs fed with quercetin-added pancreatic prescription diet.** TP, total protein; ALB, albumin; ALT, alanine aminotransferase; AST, aspartate aminotransferase; ALP, alkaline phosphatase; TG, triglyceride; TC, total cholesterol; α-AMY, α-amylase; CK, creatine kinase; CREA, creatinine; BUN, blood urea nitrogen; $Ca^{2+}$, calcium ion; $K^+$, potassium ion; $Na^+$, sodium ion. *Represents a P value less than 0.05, n.s. represents a P value greater than 0.05 (not statistically significant), compared with the control diet group. Statistical analysis was carried out by Student's two-tailed t test.

quercetin-added pancreatic prescription diet could promote the pancreatic function, and had no adverse effects on the function of liver, kidney and heart.

### 3.4. Effects of quercetin-added pancreatic prescription diet on metabolic characteristic in dogs

To analyze the effects of pancreatic prescription diet on the metabolic status of dogs, plasma samples from control-diet and pancreatic prescription-diet dogs at 0 w and 8 w were taken for untargeted metabonomics detection. A total of 1293 plasma metabolites were identified in the untargeted metabolome, of which 675 (52.2%) positive and 433 (33.5%) negative metabolites had clear chemical taxonomy information. By quantitative and proportional analysis, it was found that eight main metabolites were abundant in canine plasma, including lipid and lipid molecules (positive:193 species, 28.5%; negative: 200 species, 46.2%), organic acids and their derivatives (positive:163 species, 24.1%; negative: 88 species, 20.3%), organo-heterocyclic compounds (positive: 105 species, 15.6%; negative: 35 species, 8.08%) and others (Fig 3A and B). Principal component analysis (PCA) was used to analyze the overall pattern of plasma metabolic profiles of control diet dogs and pancreatic prescription diet dogs at 0 w and 8 w. It was found that the plasma metabolic patterns in the two groups were highly overlapped at 0 w, after two weeks of adaptive feeding. The overall metabolic characteristics of the control diet dogs were not significantly changed after eight-week feeding (Fig 3C and D). The above results indicate that all dogs showed no significant differences in metabolite

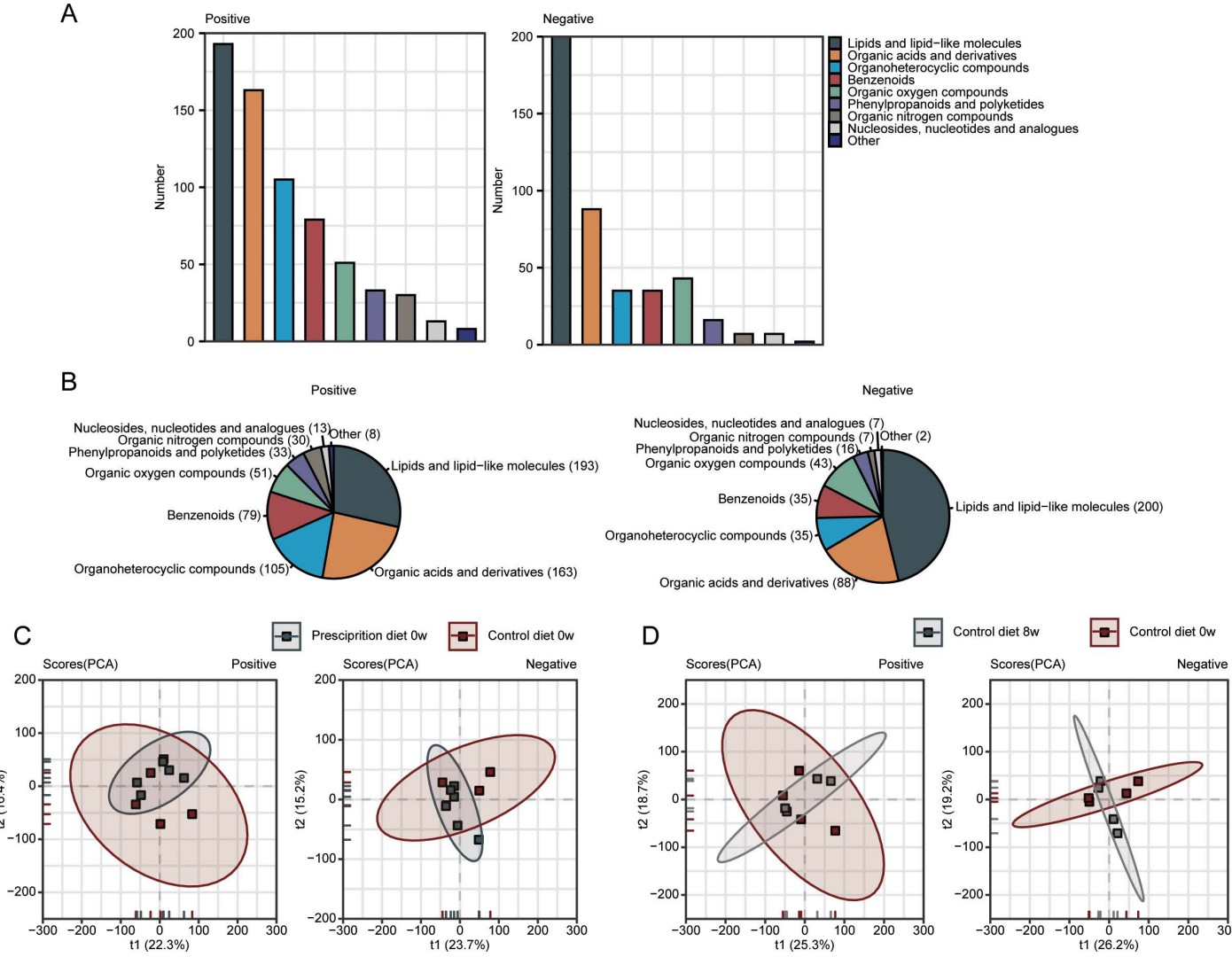

**Fig 3. Untargeted metabolomics analysis of canine plasma.** (A and B) Statistics of all identified metabolites according to the attribution information of chemical taxonomy. The specific number (A) and the proportion (B) of each metabolite superclass in plasma samples are presented, respectively. (C and D) Principal component analysis (PCA) of plasma samples in control diet group and prescription diet group at 0w (C), and in control diet group at 0w and 8w (D) in positive (left) and negative (right) ion modes.

levels prior to enrollment, and the control diet had minimal impact on their metabolite composition.

The confidence interval distributions of plasma samples were clearly separated in pancreatic prescription-diet group at 0 w and 8 w (Fig 4A), and in pancreatic prescription-diet group and control-diet group at 8 w (Fig 4B), indicating there was a significant change in the overall abundance of the blood metabolites. Comparing the total abundance and difference of eight main metabolites in plasma between the two groups at 0 w and 8 w, the results showed that the abundance of organic nitrogen compounds in the pancreatic prescription diet group was decreased, while the abundance of organic oxygen compounds, phenylpropane and polyke-ones was increased at 8 w (Fig 4C).

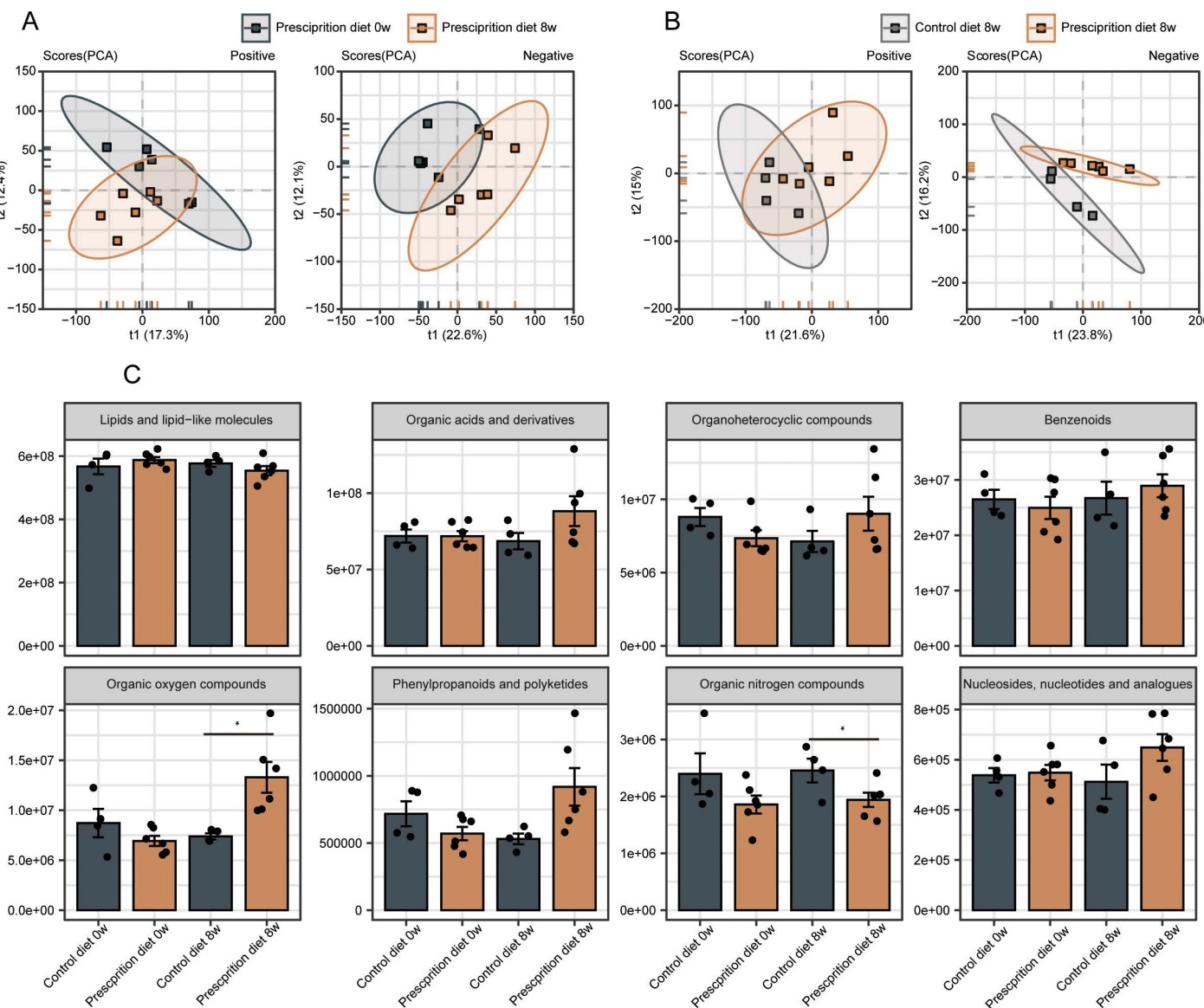

**Fig 4. The metabolic characteristic of plasma in dogs fed with quercetin-added pancreatic prescription diet.** (A and B) Principal component analysis (PCA) of plasma samples in prescription diet group at 0w and 8w (A), and in control diet group and prescription diet group at 8w (B) in positive (left) and negative (right) ion modes. (C) The comparison of the total abundance and difference of main metabolites in plasma between control diet group and prescription diet group at 0w and 8w. *Represents a P value less than 0.05, n.s. represents a P value greater than 0.05 (not statistically significant), compared with the control diet group. Statistical analysis was carried out by Student's two-tailed t test.

## 3.5. Quercetin-added pancreatic prescription diet regulates pancreatic function through pentose phosphate pathway and others

After PCA analysis of plasma metabolic profiles in control and pancreatic prescription diet dogs at 0 w and 8 w, the OPLS-DA model was applied to screen the characteristic differential metabolites. The results of model replacement test showed that the model interpretation rate (R2) and prediction ability (Q2) of the random model gradually decreased with the decrease of replacement retention, indicating that the model has good robustness and can be used for feature selection and modeling. Subsequently, model VIP > 1 and threshold P < 0.05 of

inter-group T-test were used to screen metabolites with characteristic differences in pancreatic prescription-diet group at 0 w and 8 w (Fig 5A and B). The same model was used to screen the characteristic differential metabolites in pancreatic prescription-diet group and control-diet group at 8 w (Fig 5C and D). The KEGG annotation was performed for the metabolites with different abundance to calculate the enrichment degree of each pathway. The results showed that the metabolites with different abundance were mainly concentrated in ABC transporter, arginine and proline metabolism, tyrosine metabolism, unsaturated fatty acid synthesis and other pathways (Fig 5E and F). Combined with the overall change trend of metabolites in each enrichment pathway in the pancreatic prescription-diet group at 0 w and 8 w, the absolute value of differential abundance scores was set as greater than two times threshold. it was found that the abundance of metabolites in ABC transporters, arginine and proline metabolism, tyrosine metabolism, primary bile acid biosynthesis, pentose phosphate pathway, pentose and glucuronate interconversions, and alanine, aspartate and glutamate metabolism were increased. While unsaturated fatty acids were decreased (Fig 5G). The changes of key differential metabolites in the above significantly regulated metabolic pathways were shown in Fig 5H.

## 4. Discussion

The canine pancreatitis is mainly caused by unhealthy diet habits. Obese dogs are more prone to pancreatitis, frequently after eating a lot of fatty food [14,15]. The occurrence and development of canine pancreatitis are associated with the pathological mechanisms of pancreatic microcirculation disorder, cascade effect of cytokines, systemic inflammatory response, pancreatic cell apoptosis and intestinal bacterial migration [16]. The dogs with pancreatitis are generally displayed depressed, anorexia and vomiting. The treatment of pancreatitis should be considered to remove the cause, maintain fluid and electrolyte balance and other support therapy. Treatment for complications could promote recovery of affected dogs [2]. The pancreatic prescription food adding edible Chinese medicine ingredients to enhance pancreatic function effectively ameliorate pancreatitis.

Previous studies indicated that prolonged consumption of diets rich in fats may trigger pancreatitis [17–19]. The digest function was found to be damaged in pancreatitis-attacked patients and animals. The pancreatic prescription diet with reasonable ratio of various nutritional composition, such as low fat and highly digestible, is a new adjuvant therapying approach in clinical. In the present study, we independently developed a pancreatic prescription food with quercetin supplement for dogs. the formulated pancreatic prescription food of low fat and highly digestible is conformed to the nutrition demand for pancreatitis-attacked dogs. During the experiment, all dogs could eat up the food they received and showed no abnormality in daily observation, suggestion the quercetin-added pancreatic prescription diet is a well tasting and safe food. In consideration of the principle of minimal animal use for animal welfare, 4 dogs in the control group and 6 dogs in the experimental group were used in this study. Although individual differences within groups were minimal, the limited sample size in this study may still affect the statistical power of the results.

After 8-week feeding, the quercetin-added pancreatic prescription diet group showed no changes in the body weight, fasting blood glucose, BCS, the indexes of whole blood program of red blood cells, white blood cells and platelets, and most blood biochemical indexes. These results indicated that the quercetin-added pancreatic prescription had no adverse effects on the physiological condition, and the function of liver, kidney and heart. Whereas dogs treated with quercetin-added pancreatic prescription diet showed a significant decrease in plasma lipase levels, which is a relative specific indicator of pancreatic function and a key feature of pancreatic function recovery [20]. Pancreatic lipase breaks down the majority of dietary

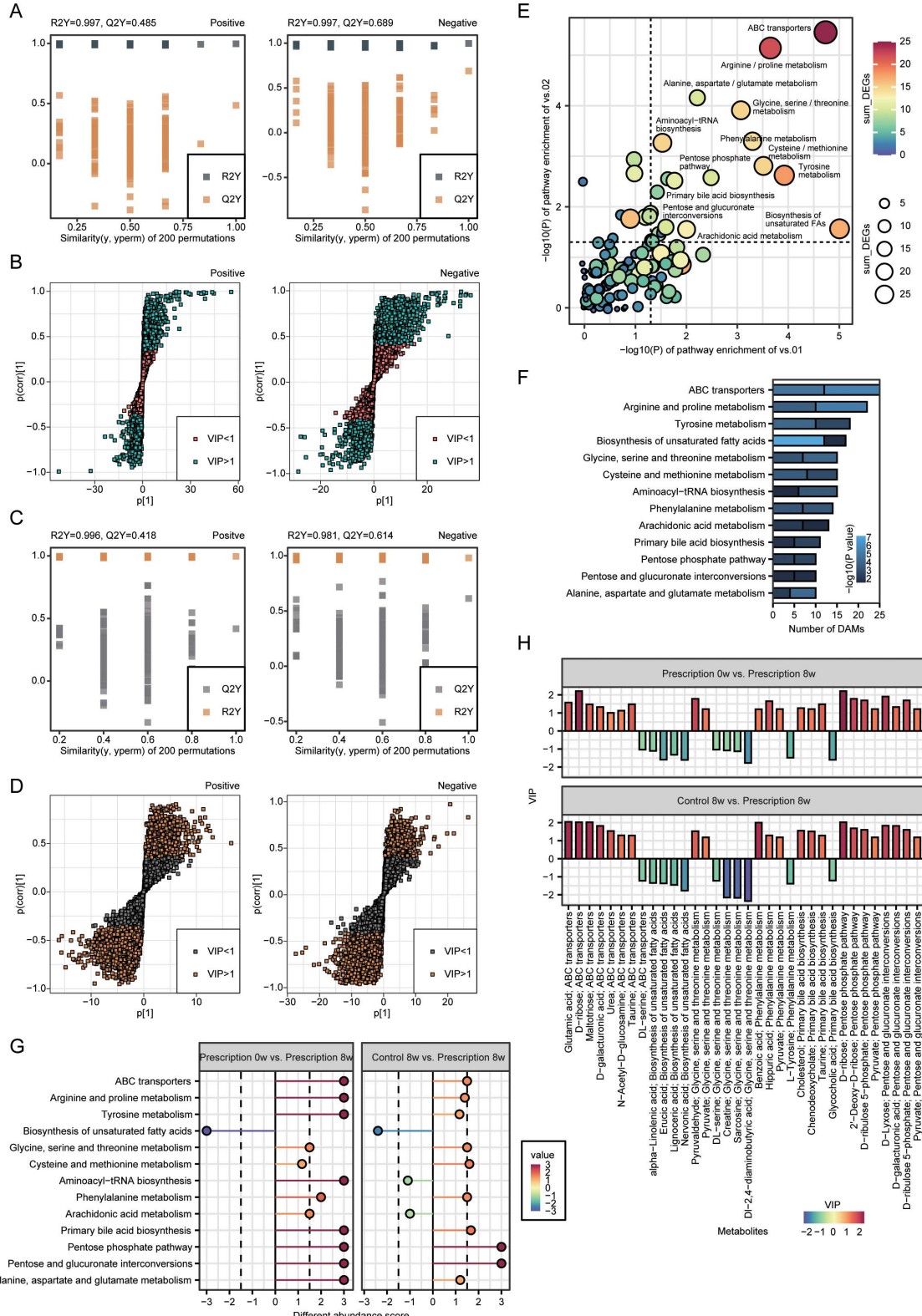

**Fig 5. Differential abundance metabolites screening and downstream pathway analysis of quercetin-added pancreatic prescription diet.** (A and B) Orthogonal partial least squares discriminant analysis (OPLS-DA) model of canine plasma samples in the prescription diet group at 0w and 8w (A), and screening of differential metabolites based on the variable importance for the projection (VIP) in the OPLS-DA model (B) in positive (left) and negative (right) ion modes. Significant difference was defined

as VIP > 1 and P < 0.05. (C and D) OPLS-DA of canine plasma samples in the control diet group and prescription diet group at 8w (C), and screening of differential metabolites based on the VIP in the OPLS-DA model (D) in positive (left) and negative (right) ion modes. Significant difference was defined as VIP > 1 and P < 0.05. (E–G) Kyoto Encyclopedia of Genes and Genomes (KEGG) pathway enrichment analysis of differential metabolites in plasma of prescription-diet dogs at 0w and 8w, and of control-diet and prescription-diet dogs at 8w. (H) The endogenous differential metabolites with the highest fold change of expression after merging positive and negative ion patterns.

triacylglycerols in the gastrointestinal tract, thereby significantly contributing to the absorption of lipids, inhibiting the lipid metabolism-related inflammatory cascades [21].

The increased abundance of pentose phosphate pathway in dogs with the quercetin-added pancreatic prescription diet is associated with the ability of anti-inflammation. The pentose phosphate pathway converts glucose-6-phosphate into pentoses and generates ribose-5-phosphate and NADPH thereby governing anabolic biosynthesis and redox homeostasis, generate fuels NOX2 to reduce reactive oxygen species which ameliorate the inflammatory response [22], the non-oxidative pentose phosphate pathway was reported to regulate regulatory T function to prevent autoimmunity [23]. The abundance of metabolites in ABC transporters in quercetin-added pancreatic prescription diet group were increased. ABC transporters reduced inflammatory signaling pathways in atherosclerosis [24], and reglated the development and function of different T cell populations [25]. ABC transporters are one of the better known mechanisms of drug resistance in cancer [26]. Chen et al. reported that quercetin could inhibit the expression levels of FZD7, ABC transporters (ABCB1, ABCC1 and ABCC2) and beta-catenin in hepatocellular carcinoma cells [27]. Quercetin-added pancreatic prescription diet in this diet played a role in pancreatic protection probably through modulating metabolic homeostasis by promoting ABC transporters and pentose phosphate pathway.

Natural plant additives were often added into diets as functional active substance. Quercetin, a flavonoid compound with various biological activities, is widely found in a variety of plants (ginkgo, notoginseng, buckwheat, elderberry, guava, etc.) and food (such as asparagus, cabbage, tea, grapes, apples, etc.). Quercetin has a wide range of biological effects. Quercetin could inhibit the growth, migration, and invasion and induced apoptosis of pancreatic cancer cells by antagonizing SHH and TGF-β/Smad signaling pathways [28]. Studies showed that quercetin down-regulated the expression and activation of NF-κB, a key inflammatory factor, in the progression of pancreatitis [2]. Previous studies reported that quercetin intervention exerted therapeutic effects on pancreatitis, intestinal injury and inflammation by inhibiting the activation of TLR4/MyD88/P38 MAPK pathway by up-regulating miR-216b and endoplasmic reticulum stress [29,30]. The pentose phosphate pathway is divided into branches that produce the nucleotide precursor ribose-5-phosphate and the cofactor NADPH, which serves as a reducing agent during the biosynthesis of lipids and amino acids [31]. Quercetin could modulate NADPH oxidase-derived $O_2^-$ production in macrophages, suppress NADPH oxidase-dependent oxidative stress to exert anti-oxidant and anti-inflammatory properties [32]. In this study, quercetin-added pancreatic prescription diet promoted pancreatic and maintain metabolic homeostasis, properly via restraining NADPH oxidase-dependent oxidative stress related to pentose phosphate pathway. However, the regulated mechanism needs further research. Whatever, the results of the present study provide evidence for treatment strategy for human pancreatitis.

Quercetin, as a common feed plant additive, is widely used in the development of functional fodder. Quercetin supplemented to the diet can improve immune function and the quality of egg in aging hens, regulate gut microbial metabolism so as to attenuate diarrhea in piglets [33–35]. In the present study, dogs feeding with quercetin-added pancreatic prescription displayed improved the pancreatic function during long-term feeding, suggesting that the quercetin-added prescription diet has a good prospect for development.

## 5. Conclusion

In conclusion, quercetin-added pancreatic prescription diet can promote pancreatic function and metabolic homeostasis via increasing pentose phosphate pathway metabolism. Our study provides new insight into molecular mechanisms of quercetin-added pancreatic prescription on therapying pancreatitis.

## Author contributions

**Conceptualization:** Xiao-Wan Liu, Zhouxiang Wang.

**Data curation:** Li Xu.

**Formal analysis:** Yaohui Zhang, Zhouxiang Wang.

**Investigation:** Jia-Bao Xing, Zhouxiang Wang, Xin Zhang.

**Methodology:** Jia-Bao Xing, Xin Zhang, Ming-Xing Ding.

**Resources:** Zhili Qi, Yi Ding, Xin Zhang, Xiao-Jing Zhang.

**Software:** Zhouxiang Wang.

**Supervision:** Juan Wan, Manli Hu, Zhili Qi, Yi Ding, Xiao-Jing Zhang.

**Validation:** Jia-Bao Xing, Manli Hu, Yun Chen.

**Visualization:** Yun Chen, Ming-Xing Ding.

**Writing – original draft:** Yaohui Zhang, Li Xu, Xiao-Wan Liu.

**Writing – review & editing:** Yaohui Zhang, Li Xu, Xiao-Wan Liu.

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
