## [Decision Letter · Decision Letter 0]

17 Sep 2024

PONE-D-24-24297Exploring the Effects of Quercetin-added Pancreatic Diet on Metabolic Homeostasis in Dogs via MetabolomicsPLOS ONE

Dear Dr. Wan,

Thank you for submitting your manuscript to PLOS ONE. After careful consideration, we feel that it has merit but does not fully meet PLOS ONE’s publication criteria as it currently stands. Therefore, we invite you to submit a revised version of the manuscript that addresses the points raised during the review process. Please submit your revised manuscript by Nov 01 2024 11:59PM. If you will need more time than this to complete your revisions, please reply to this message or contact the journal office at plosone@plos.org . Please include the following items when submitting your revised manuscript:

We look forward to receiving your revised manuscript.

Kind regards,

Ahmed E. Abdel Moneim

Academic Editor

PLOS ONE

Journal Requirements:

2.  Please provide additional details regarding participant consent (Owners of the dog). In the ethics statement in the Methods and online submission information, please ensure that you have specified (1) whether consent was informed and (2) what type you obtained (for instance, written or verbal, and if verbal, how it was documented and witnessed).

3. To comply with PLOS ONE submissions requirements, in your Methods section, please provide additional information regarding the experiments involving animals and ensure you have included details on (1) methods of sacrifice, (2) methods of anesthesia and/or analgesia, and (3) efforts to alleviate suffering.

“This work was supported by the National Science Foundation of China (Grant No. 82170595 and 81970070 to X-JZ), the Henan Charity General Federation-Hepatobiliary Foundation of Henan Charity General Federation (Grant No. GDXZ2022008 to XZ, GDXZ2022009 to WJ), Huazhong Agricultural University-Agricultural Genomics Institute at Shenzhen, Cooperation Fund (No. SZYJY2022008 to ZQ), National special fund for research and development (No. YFD051009 to M-XD).

Gannan Innovation and Transformation Medical Research Institute, 4# building, Phase II, high level talents  Park, Zhanggong high-tech zone, Ganzhou City, Jiangxi Province.”

6. Thank you for stating the following in the Competing Interests section:

“We would like to submit the enclosed manuscript entitled “Exploring the Effects of Quercetin-added Pancreatic Diet on Metabolic Homeostasis in Dogs via Metabolomics”. No conflict of interest exits in the submission of this manuscript, and manuscript is approved by all authors for publication. I would like to declare on behalf of my co-authors that the work described was original research that has not been published previously, and not under consideration for publication elsewhere, in whole or in part. All the authors listed have approved the manuscript that is enclosed.”

7. We note that your Data Availability Statement is currently as follows: [All relevant data are within the manuscript and its Supporting Information files.]

Reviewers' comments:

Reviewer's Responses to Questions

**Comments to the Author**

1. Is the manuscript technically sound, and do the data support the conclusions?

Reviewer #1: Yes

Reviewer #2: Yes

2. Has the statistical analysis been performed appropriately and rigorously? 

Reviewer #1: Yes

Reviewer #2: Yes

3. Have the authors made all data underlying the findings in their manuscript fully available?

Reviewer #1: Yes

Reviewer #2: Yes

4. Is the manuscript presented in an intelligible fashion and written in standard English?

Reviewer #1: Yes

Reviewer #2: Yes

5. Review Comments to the Author

Reviewer #1: The aim and conclusion have been aligned. However, it could be a better paper with some improvements in some areas, such as enhancing clarity in the abstract method, strengthening the argumentation by considering adding more recent studies, data points, or supporting arguments where the paper feels underdeveloped, and minor detail improvements such as improving the flow and revising transitions between sections to ensure the paper flows smoothly.

Reviewer #2: Comments and Suggestions for Authors

The study titled "Exploring the Effects of Quercetin-Enhanced Pancreatic Diet on Metabolic Homeostasis in Dogs Using Metabolomics," authored by Yaohui Zhang et al., investigates the role of a quercetin-enriched pancreatic prescription diet in regulating metabolic homeostasis in dogs, with a particular focus on fatty acid, amino acid, and bile acid metabolism. Utilizing advanced techniques such as untargeted metabolomics and bioinformatics analysis, the research examines the diet's impact on key metabolic pathways. The findings suggest that long-term administration of a quercetin-enhanced diet may yield significant and lasting benefits for pancreatic health in dogs. This study highlights the diet's potential to support pancreatic function and promote metabolic homeostasis, offering valuable insights into the therapeutic potential of quercetin for managing pancreatitis. The manuscript is well-written. The reviewer has only some minor concerns as follows:

1. In the Materials and Methods section, the study was conducted with a limited sample size of 10 dogs, comprising 6 in the experimental group and 4 in the control group. This small sample size may limit the generalizability and statistical power of the results. The authors may wish to discuss additional factors that influenced the decision to use a smaller sample size, such as ethical considerations related to animal welfare or other relevant constraints.

2. In this study, authors mentioned in abstract and that untargeted metabolomics analysis revealed that quercetin activates ABC transport and arginine/proline pathways, suggesting potential benefits for pancreatitis in large animals. However, author may need to discuss more detail or analyze current metabolomics data to clarify the role of ABC transport on promoted pancreatic function and sustain metabolic homeostasis after animal fed quercetin added food.

3. In Figure 3, the Principal Component Analysis (PCA) revealed overlapping confidence intervals in some instances, suggesting that the metabolic profiles between groups may not be as distinct as initially claimed. The authors might consider discussing this observation more detail in the Results or Discussion section.

4. While the study demonstrates that quercetin influences metabolic pathways such as ABC transporters and the pentose phosphate pathway, the authors could use the current data to provide a more detailed mechanistic explanation of how quercetin exerts these effects. The absence of this analysis leaves an incomplete understanding of quercetin's role. The authors may wish to address this in the Discussion section.

6. PLOS authors have the option to publish the peer review history of their article (what does this mean? ). If published, this will include your full peer review and any attached files.

**Do you want your identity to be public for this peer review?** For information about this choice, including consent withdrawal, please see our Privacy Policy .

Reviewer #1: **Yes: ** Sigid Prabowo

Reviewer #2: No

---

## [Author Response · Author response to Decision Letter 0]

15 Dec 2024

Dear Editors and Reviewer,

Thank you very much for your critical comments and thoughtful suggestions. We have made careful modification on the original manuscript. All changes made to the text are in red in the revised manuscript so that they made be easily identified. All the questions are answered below.

Response to Reviewer 1:

Q1. In the Materials and Methods section, the study was conducted with a limited sample size of 10 dogs, comprising 6 in the experimental group and 4 in the control group. This small sample size may limit the generalizability and statistical power of the

results. The authors may wish to discuss additional factors that influenced the decision to use a smaller sample size, such as ethical considerations related to animal welfare or other relevant constraints.

Response 1: We acknowledge this good suggestion. We added “In consideration of the principle of minimal animal use for animal welfare, 4 dogs in the control group and 6 dogs in the experimental group were used in this study. Although individual differences within groups were minimal, the limited sample size in this study may still affect the statistical power of the results.” in Discussion section in Line 316-319.

Q2. In this study, authors mentioned in abstract and that untargeted metabolomics

analysis revealed that quercetin activates ABC transport and arginine/proline

pathways, suggesting potential benefits for pancreatitis in large animals. However,

author may need to discuss more detail or analyze current metabolomics data to clarify

the role of ABC transport on promoted pancreatic function and sustain metabolic homeostasis after animal fed quercetin added food.

Response 2: Thanks for your comments. The abundance of metabolites in ABC transporters in quercetin-added pancreatic prescription diet group were increased. ABC transporters reduced inflammatory signaling pathways in atherosclerosis (Fitzgerald ML, Mujawar Z, Tamehiro N. ABC transporters, atherosclerosis and inflammation. Atherosclerosis. 2010 Aug;211(2):361-70. doi: 10.1016/j.atherosclerosis.2010.01.011), and reglated the development and function of different T cell populations(Thurm C, Schraven B, Kahlfuss S. ABC Transporters in T Cell-Mediated Physiological and Pathological Immune Responses. Int J Mol Sci. 2021 Aug 25;22(17):9186. doi: 10.3390/ijms22179186). ABC transporters are one of the better known mechanisms of drug resistance in cancer (Gu, J., Huang, W., Wang, X. et al. Hsa-miR-3178/RhoB/PI3K/Akt, a novel signaling pathway regulates ABC transporters to reverse gemcitabine resistance in pancreatic cancer. Mol Cancer 21, 112 (2022). https://doi.org/10.1186/s12943-022-01587-9; El-Mahdy HA, El-Husseiny AA, Kandil YI, Gamal El-Din AM. Diltiazem potentiates the cytotoxicity of gemcitabine and 5-fluorouracil in PANC-1 human pancreatic cancer cells through inhibition of P-glycoprotein. Life Sci. 2020;262:118518). Chen et al. reported that quercetin could inhibit the expression levels of FZD7, ABC transporters (ABCB1, ABCC1 and ABCC2) and beta-catenin in hepatocellular carcinoma cells (Chen Z, Huang C, Ma T, Jiang L, Tang L, Shi T, Zhang S, Zhang L, Zhu P, Li J, Shen A. Reversal effect of quercetin on multidrug resistance via FZD7/β-catenin pathway in hepatocellular carcinoma cells. Phytomedicine. 2018 Apr 1;43:37-45. doi: 10.1016/j.phymed.2018.03.040). Quercetin-added pancreatic prescription diet in this diet played a role in pancreatic protection probably through modulating metabolic homeostasis by promoting ABC transporters and pentose phosphate pathway.

The related content has been added to the Discussion section in Line 368-370.

Q3. In Figure 3, the Principal Component Analysis (PCA) revealed overlapping

confidence intervals in some instances, suggesting that the metabolic profiles between

groups may not be as distinct as initially claimed. The authors might consider discussing this observation more detail in the Results or Discussion section.

Response 3: Thanks for your suggestions. We have added “The above results indicate that all dogs showed no significant differences in metabolite levels prior to enrollment, and the control diet had minimal impact on their metabolite composition” in the Result section in Line 273-275.

Q4. While the study demonstrates that quercetin influences metabolic pathways such as ABC transporters and the pentose phosphate pathway, the authors could use the

current data to provide a more detailed mechanistic explanation of how quercetin

exerts these effects. The absence of this analysis leaves an incomplete understanding of quercetin's role. The authors may wish to address this in the Discussion section.

Response 4: We sincerely thank the reviewer for good suggestion. The pentose phosphate pathway is divided into branches that produce the nucleotide precursor ribose-5-phosphate and the cofactor NADPH, which serves as a reducing agent during the biosynthesis of lipids and amino acids (Stincone A, Prigione A, Cramer T, Wamelink MM, Campbell K, Cheung E, Olin-Sandoval V, Grüning NM, Krüger A, Tauqeer Alam M, Keller MA, Breitenbach M, Brindle KM, Rabinowitz JD, Ralser M. The return of metabolism: biochemistry and physiology of the pentose phosphate pathway. Biol Rev Camb Philos Soc. 2015 Aug;90(3):927-63. doi: 10.1111/brv.12140). Quercetin could modulate NADPH oxidase-derived O2.- production in macrophages, suppress NADPH oxidase-dependent oxidative stress to exert anti-oxidant and anti-inflammatory properties. In this study, quercetin-added pancreatic prescription diet promoted pancreatic and maintain metabolic homeostasis, properly via restraining NADPH oxidase-dependent oxidative stress related to pentose phosphate pathway.

The related content has been added to the Discussion section in Line 383-391.

Response to Reviewer 2:

Q1: Initiate with the information on canine physical examination results is strongly advocated.

Response 1: We sincerely appreciate the valuable comments. We have added the results of the body weight, fasting blood glucose, BCS, the indexes of whole blood program, and the blood biochemical indexes in the Abstract. Please check the details in Line 53-61.

Q2: For better paper structure, the methods to measure variables are categorized according to the following:

1. Canine physical examination,

2. Complete blood count and serum biochemical tests

3. Untargeted metabolomics analysis

Aligned with the information in the abstract.

Response 2: Thanks for your critical comments. According to your suggestion, we have adjusted the method structure in the revised manuscript.

Q3: refocused on the measured variables:

1. Canine physical examination,

2. Complete blood count and serum biochemical tests

3. Untargeted metabolomics analysis

Then, compare the findings data with references.

Response 3: As suggested by the reviewer, we have added the discussion about canine physical indexes, the indexes of whole blood program, and the blood biochemical indexes, and compared our data with references. The related content are in the Discussion section in Line 361-370.

---

## [Decision Letter · Decision Letter 1]

12 Jan 2025

Exploring the Effects of Quercetin-added Pancreatic Diet on Metabolic Homeostasis in Dogs via Metabolomics

PONE-D-24-24297R1

Dear Dr. Wan,

We’re pleased to inform you that your manuscript has been judged scientifically suitable for publication and will be formally accepted for publication once it meets all outstanding technical requirements.

Kind regards,

Ahmed E. Abdel Moneim

Academic Editor

PLOS ONE

Additional Editor Comments (optional):

Reviewers' comments:

Reviewer's Responses to Questions

**Comments to the Author**

1. If the authors have adequately addressed your comments raised in a previous round of review and you feel that this manuscript is now acceptable for publication, you may indicate that here to bypass the “Comments to the Author” section, enter your conflict of interest statement in the “Confidential to Editor” section, and submit your "Accept" recommendation.

Reviewer #1: All comments have been addressed

Reviewer #2: All comments have been addressed

2. Is the manuscript technically sound, and do the data support the conclusions?

Reviewer #1: Yes

Reviewer #2: Yes

3. Has the statistical analysis been performed appropriately and rigorously? 

Reviewer #1: Yes

Reviewer #2: Yes

4. Have the authors made all data underlying the findings in their manuscript fully available?

Reviewer #1: Yes

Reviewer #2: Yes

5. Is the manuscript presented in an intelligible fashion and written in standard English?

Reviewer #1: Yes

Reviewer #2: Yes

6. Review Comments to the Author

Reviewer #1: The reviewer's comments have been thoroughly addressed, ensuring that each point raised has been carefully analyzed and responded to with precision. Specific areas of concern, such as methodological clarifications, data interpretation, or theoretical frameworks, were revisited to ensure alignment with the reviewer's expectations and to enhance the overall quality of the work. Detailed explanations, supported by relevant references or additional data where necessary, were provided to substantiate the responses. Any ambiguities or inconsistencies identified by the reviewer were resolved through revisions, ensuring coherence and rigor in the document. Moreover, all suggested improvements, including formatting adjustments or content refinements, were incorporated to meet the required standards. This comprehensive approach not only strengthens the validity of the work but also demonstrates a commitment to maintaining high academic and professional standards.

Reviewer #2: The study titled "Exploring the Effects of Quercetin-Enhanced Pancreatic Diet on Metabolic Homeostasis in Dogs Using Metabolomics," authored by Yaohui Zhang et al., investigates the role of a quercetin-enriched pancreatic prescription diet in regulating metabolic homeostasis in dogs, with a particular focus on fatty acid, amino acid, and bile acid metabolism. Utilizing advanced techniques such as untargeted metabolomics and bioinformatics analysis, the research examines the diet's impact on key metabolic pathways. The findings suggest that long-term administration of a quercetin-enhanced diet may yield significant and lasting benefits for pancreatic health in dogs. This study highlights the diet's potential to support pancreatic function and promote metabolic homeostasis, offering valuable insights into the therapeutic potential of quercetin for managing pancreatitis

7. PLOS authors have the option to publish the peer review history of their article (what does this mean? ). If published, this will include your full peer review and any attached files.

**Do you want your identity to be public for this peer review?** For information about this choice, including consent withdrawal, please see our Privacy Policy .

Reviewer #1: **Yes: ** Sigid Prabowo

Reviewer #2: No

---

## [Editor Report · Acceptance letter]

PONE-D-24-24297R1

PLOS ONE

Dear Dr. Wan,

I'm pleased to inform you that your manuscript has been deemed suitable for publication in PLOS ONE. Congratulations! Your manuscript is now being handed over to our production team.

Kind regards,

on behalf of

Dr. Ahmed E. Abdel Moneim

Academic Editor

PLOS ONE